# Long-Term Effects of Breast Cancer Therapy and Care: Calm after the Storm?

**DOI:** 10.3390/jcm11237239

**Published:** 2022-12-06

**Authors:** Chiara Tommasi, Rita Balsano, Matilde Corianò, Benedetta Pellegrino, Giorgio Saba, Fabio Bardanzellu, Nerina Denaro, Matteo Ramundo, Ilaria Toma, Alessandro Fusaro, Serafina Martella, Marco Maria Aiello, Mario Scartozzi, Antonino Musolino, Cinzia Solinas

**Affiliations:** 1Medical Oncology and Breast Unit, University Hospital of Parma, 43126 Parma, Italy; 2Department of Medicine and Surgery, University of Parma, 43126 Parma, Italy; 3GOIRC (Gruppo Oncologico Italiano di Ricerca Clinica), 43126 Parma, Italy; 4Medical Oncology Department, University of Cagliari, 09042 Cagliari, Italy; 5Medical Oncology, Fondazione Ca’ Grande Ospedale Maggiore Policlinico, 20122 Milan, Italy; 6Unità Operativa Complessa di Oncologia, Azienda Ospedaliera Cardinale G. Panico, 73039 Tricase, Italy; 7LILT Valle d’Aosta, 11100 Aosta, Italy; 8Medical Oncology, University Hospital Policlinico San Marco, 95123 Catania, Italy

**Keywords:** breast cancer, breast cancer survivors, quality of life, side effects, toxicity, psychological impact

## Abstract

Breast cancer is still a lethal disease and the leading cause of death in women, undermining patients’ survival and quality of life. Modern techniques of surgery and radiotherapy allow for the obtaining of good results in terms of survival, however they cause long-term side effects that persist over time, such as lymphedema and neuropathy. Similarly, the advent of new therapies such as endocrine therapy revolutionized breast cancer outcomes, but side effects are still present even in years of follow-up after cure. Besides the side effects of medical and surgical therapy, breast cancer is a real disruption in patients’ lives considering quality of life-related aspects such as the distortion of body image, the psychological consequences of the diagnosis, and the impact on family dynamics. Therefore, the doctor-patient relationship is central to providing the best support both during treatment and afterwards. The aim of this review is to summarize the consequences of medical and surgical treatment on breast cancer patients and to emphasize the importance of early prevention of side effects to improve patients’ quality of life.

## 1. Introduction

Breast cancer (BC) is the most common tumor and the leading cause of death in women [1]. In 2020, 2.3 million women had a diagnosis of breast cancer, and this accounted for 685,000 of deaths worldwide [1]. BC has emerged as the most prevalent cancer since the end of 2020: in fact, 7.8 million women received a diagnosis in the past 5 years [1]. Incidence rates are lower in regions of Africa and Central Asia; while they are elevated in Australia, Europe and North America. Notably, despite their high incidence rates, some European, North American and Australian countries have among the lowest mortality rates [2]. The BC incidence variations across countries can be related with differences in the distribution of risk factors (e.g., reproductive factors or obesity) and with the degree of development of screening programs [2]. The urvival of BC patients changes according to stage, molecular subtype and histology of the disease: in Europe, women with early BC have a 5-years survival rate of 96% compared with 38% with metastatic BC at diagnosis [3]. In developed countries, the life-expectation of patients with BC has improved in the last years and this is due to the implementation of screening programs, leading to earlier diagnosis, to the improvements in tumor molecular characterization, and innovative and tailored treatments [4].

Nevertheless, the increased number of BC survivors has uncovered new problems related to BC treatments which have a significant impact on women’s physical, psychosocial and emotional health [5]. In fact, BC survivors usually deal with a wide range of symptoms related to surgery, chemotherapy (CT), radiotherapy (RT) or other adjuvant treatments. The management of these conditions represents a currently debated issue that requires improvements and that aims to define the best follow-up time schedule, the prevention of the early onset of treatment-related side effects, but also the management of long-term toxicities with the relative psychological consequences.

The aim of this review is to focus the attention on the main side effects of BC treatments and their psychological and physical impact on patients and to identify some possible solutions.

## 2. Side Effects of Local Treatments

### 2.1. Surgery

#### 2.1.1. Breast Surgery

Surgery has been known as the procedure of choice for BC management since 1700. Currently, breast-conserving surgery or mastectomy are performed in selected cases with specific clinical criteria [6].

From a technical point of view, breast reconstruction can include either autologous flaps or implants. Instead, women preferring no reconstruction often choose a prothesis [5]. The best outcomes are found with autologous flaps from the abdomen or gluteus [7], while protheses often cause social discomfort and pain [8].

However, either mastectomy or breast-conserving surgery could have several long-term effects. It is often reported that, after mastectomy, women report shoulder discomfort, and even “rotator cuff disease” can occur due to the changes apported by surgery on the joints [5,9].

Furthermore, up to 80% of women are diagnosed with “phantom breast pain” after mastectomy, which is defined as a complex of symptoms such as itch and pain in the amputated breast [10]. Risk factors for the “phantom breast syndrome” are severe acute post-operative pain, increased analgesic use, and history of neuropathic pain disorders and immediate breast reconstruction after surgery [5]. The treatment of this syndrome includes oral medications such as antidepressants, anticonvulsants, opioids, cannabinoids, and topical anesthetics, while other neuromodulation techniques (i.e., motor cortex and spinal cord stimulation) are under study [10].

Regarding the consequences to the surrounding issues, breast surgery apports damage to blood vessels, nerves and lymph-vessels, causing fibrosis and gait disorders, especially after unilateral mastectomy [9], but also lymphedema of the upper limb [11]. The most common problems with shoulder movements are related to internal rotation and extension: following a strict exercise program compared to usual care drastically improved the upper limb function in patients at risk of arm disability after surgery [12].

#### 2.1.2. Lymphedema

Lymphedema is a chronic and progressive condition characterized by the accumulation of fluid in the interstitial tissues of the arm, shoulder and neck [13].

Of course, the most important risk factor is axillary lymph node dissection (ALND) [13], which leads to a four times higher risk of lymphedema [14]; therefore, the currently used sentinel lymph node biopsy has decreased its incidence [15]. By the way, lymphedema can also occur as a long-term effect: in fact, it can appear even 2–5 years after surgery [14]. It affects 10% of women diagnosed with breast cancer, causing symptoms such as homolateral arm pain and numbness [14].

Other risk factors related to lymphedema development are the number of lymph nodes dissected, mastectomy, and adjuvant therapies [16]. Lymphedema has a huge impact on the quality of life of survivors [16]: it is associated with pain and reduced strength related to the feeling of heaviness; this translates into limitation of movements and reduced physical activity [16,17]. Overweight also contributes to the onset of lymphedema: women with a body mass index (BMI) > 25 had a higher risk of developing lymphedema than those with BMI < 25 [18].

Of note, the presence of lymphedema needs the right treatment and an accurate follow-up to prevent the possible negative impact on the life of BC survivors [19]. The treatments of lymphedema are usually non-surgical, such as early physiotherapy and lymphatic drainage with garments or manual drainage techniques, or, rarely, surgical approaches (i.e., microsurgical lymphatic surgery or suction lipectomy) [20].

Figure 1 summarizes the most important effects of surgery.

### 2.2. Radiotherapy

Postoperative RT is strongly recommended after BC conserving surgery [21]: whole-breast RT (WBRT) alone reduces by 15% the 10-year risk of any first recurrence (including locoregional and distant) and by 4% the 15-year risk of BC-related mortality [22]. Nevertheless, it is associated with several side effects that can persist over time.

One of the most important consequences of breast RT is the lung damage which can lead to early effects such as pneumonitis onset or later effects such as lung fibrosis [23]. The risk of lung damage is related to the various techniques of RT and to the site and dosage of irradiation: the risk is higher with the irradiation of the mammary chain [24], but a less than 20 Gy dosage is unlikely to affect the lungs [25]. However, in the last decades, new RT techniques has been invented, such as non-coplanar volumetric modulated arc therapy, which seems to reduce lung damage [26].

Another consequence due to RT could be the affection of the functionality of arms with an increased incidence of lymphedema, neuropathy and altered shoulder mobility with pain and reduced movement such as flexion and abduction [27]. They are caused by RT-induced fibrosis and vascular damage: in a self-reported morbidity trial, 90% of patients reported these symptoms after treatment [28]. In particular, “brachial plexus neuropathy” is a syndrome caused by the damage of nerves, accompanied by motorial and sensitive symptoms including hypoesthesia, hypoalgesia and muscular atrophy: fibrosis induced by RT, in fact, can involve the plexus, with potentiallyirreversible damage [29].

Another organ which could suffer from RT is the heart. Indeed, late cardiac toxicities caused by (particularly left-sided) WBRT are now recognized as rare but relevant sequelae [30]. A population-based case-control study revealed that the relative risk of major coronary events increased linearly with the mean heart dose by 7.4% per Gy [31]. With regard to mean heart dose, WBRT treatment planning should also include constraints for cardiac sub-volumes. The individual decision between sufficient protection of cardiac structures versus optimal target volume coverage remains in the physician’s hands. The breast cancer-specific mortality and patients’ cardiac risk factors must be individually weighed against the risk of radiation-induced cardiotoxicity [30].

Radiation-induced angiosarcoma (RIAS) is a rare consequence of breast radiation: the incidence of RIAS in women treated with breast-conserving surgery and adjuvant RT can vary from 0.14% to about 0.5% and it affects older women (mean age 71 years old) [32]. The oncogenic effect of ionizing radiation and the cellular repairing stimulation after tissue damage are associated with RIAS development; the latency of breast RIAS is shortened when compared with other radiation-induced sarcomas and it may develop early (at about 6 months), but also after 40 years after RT (median latency of 6 years) [32].

The awareness of long-time side-effects of RT is fundamental and should be part of the treatment of BC patients to preserve and prevent the quality of life of BC survivors. For this reason, some clinical trials (Table 1) are evaluating the impact of omission/reduction of RT in patients with low-risk BC.

### 2.3. The Coexistence with the New Body Image

Surgery, lymphadenectomy and RT have a less considered but really important emotional impact due to the coexistence with the changes of external features of the body mainly due to the local treatments; this represents one of the most important issues related to surviving BC patients.

Regarding surgery, the rate of mastectomy is currently increasing, although 90% of women with an early stage BC could undergo breast-conserving therapy (surgery and RT) [38]. This trend is particularly pronounced in young women and could be related to patients’ worries about breast symmetry, fear of recurrence, and long-term sequelae of breast-conserving therapy. In fact, many women prefer to receive extensive surgery with its higher risks rather than breast-conservative therapy in order to preserve their self-image and feel more at ease in the future [38].

In addition, upper limb lymphedema and the breast aspect after RT can affect body-image, causing a reduction in self-confidence and distress related to negative feelings [39].

The concept of body image is a complex post-treatment issue for BC survivors and the data on it in the literature are limited by the high variability of methods used to “measure” the effect of body image on patients [40].

A study found that in a sample of 1956 women with a diagnosis of BC, 38% reported lost self-confidence and 44% felt uncomfortable with their body [41]. It has been highlighted how the negative impact of BC on self-confidence persists over time, despite further surgery or reconstruction [42]. More attention to these issues is required to address the psychological health of the patients.

## 3. Side Effects of Systemic Treatments

Neoadjuvant and adjuvant CT regimens have an extensive range of toxicity that varies with respect to the agents, frequency, and administration route. They include early effects related to the half-life and mechanisms of action of drugs, but also long-term effects [43].

### 3.1. Chemotherapy-Induced Neuropathy

Chemotherapy-induced peripheral neuropathy (CIPN) is a detrimental effect reported with platinum-based regimens (e.g., carboplatin, cisplatin and oxaliplatin), taxanes (e.g., paclitaxel and docetaxel) and vinca alkaloids (e.g., vinblastine) [44]. It is manifested by progressive and often irreversible damage during the administration of antineoplastic drugs [44].

Microtubules are fundamental to the myelination of nervous fibers and are essential constituents of oligodendrocytes [45]. Taxanes and vinca alkaloids target microtubules blocking polymerization during the mitotic phase of the cell cycle [46]. Their effect on microtubules damages the small fiber sensory axons and leads to predominantly distal sensory neuropathy, which is characterized by pain, numbness, tingling, and reduced functional capacity in the extremities [46]. These drugs are milestones of BC treatment in both early and advanced settings, resulting in a high percentage of women suffering from CIPN at any grade up to 80%, with consequent high percentage of treatment discontinuation [29].

CIPN can also include cognitive impairment, also known as ‘chemofog’ or ‘chemobrain’, with a disfunction in working memory and processing speed [47]. As a demonstration, a cross-sectional study found alterations in memory and motorial functions in the 13–70% of a subgroup of BC survivors [48]. Furthermore, in another study conducted by Cavaletti et al., neurotoxicity among patients after taxane-based adjuvant CT was evaluated at different time points and persistent neurological discomfort up to 2 years after the end of therapy was reported [49]. Similarly, Ahles et al. found lower scores regarding psychomotor functioning in BC survivors treated with CT 5 years previously [50]. In addition, some researches correlate CIPN with distress and poor sleep quality, causing a worse quality of life in cancer survivors [51]. Up to 40% of cancer survivors reported sleep problems: in particular, BC survivors report poor sleep quality associated with symptoms such as anxiety, neuropathic pain and sensory neuropathy [52].

Although neurotoxicity is a well-known long-term effect and many trials have been made to try to find a prevention or a therapy for CIPN, there is no a well-established treatment. Only duloxetine, a selective serotonin reuptake inhibitor, is recommended for the treatment of neuropathic pain [53].

### 3.2. Cardiotoxicity

Treatment of BC consists of a range of different drugs that could affect cardiac function.

#### 3.2.1. Effects of Chemotherapy and Hormonal Treatments on the Cardiovascular System

Anthracyclines are a mainstay of BC therapy, in particular in the neoadjuvant and adjuvant setting, with strong evidence with regard to survival [54]. However, they exert a cytostatic effect resulting in myocardial damage through the generation of reactive oxygen species (treatment-related type I cardiac damage) [55]. Side effects of anthracyclines include arrhythmias, pericarditis, myocarditis and late-onset chronic heart failure (CHF) [56]. Cardiac damage is dose-dependent: after a cumulative dose of 400 mg/m^2^, the risk of CHF could be up to 5.1% [57].

The preventive strategies to avoid the development of Anthracyclines-related damage is the identification of higher risk patients, the promotion of health behaviours, including weight control, smoking cessation and regular aerobic exercise. The monitoring of cardiac function during treatment is required to identify possible early signs of damage [58]. Furthermore, taxanes present a dose-dependent cardiovascular risk: paclitaxel is related to a risk of bradycardia in 30% of patients. In addition, taxanes can enhance anthracyclines toxicity at high doses [59]. Because of the morbidity related with anthracyclines-related cardiotoxicity, different trial and evidence from real-world studies demonstrated the impact their omission, in particular in early HER2+ BC patients who underwent neo-/adjuvant CT [60,61,62].

Hormonal agents affect the cardiovascular system: the effects of aromatase inhibitors (AIs) on lipid levels and metabolism are variable, and in part unknown [63]. Tamoxifen reduces cholesterol and homocysteine, increasing triglycerides serum levels, a known risk factor for venous thromboembolic disease [64]. In fact, the incidence of thromboembolic events is higher in patients treated with tamoxifen rather than placebo [65].

AIs treatment has a significantly fewer thromboembolic effect and a similar incidence of ischemic cardiovascular events compared with tamoxifen [63]. Extended endocrine therapy with AIs more than 5 years significantly increased the risk of cardiotoxicity, while the risk of hypertension and hypercholesterolemia remained stable [66].

#### 3.2.2. Effects of Target Therapy on Cardiovascular System

Target therapy agents are predominantly associated with arterial hypertension, cardiac arrhythmias, and ventricular dysfunction; these adverse events are usually reversible, not related to cumulative dose, and not associated with cardiomyocytes’ structural alterations (treatment-related type II cardiac damage) [55].

Anti-HER2 antibodies are humanized antibodies binding the extracellular domain of the human epidermal growth factor receptor 2 (HER2): they are used in BC with HER2 amplification/overexpression. HER2 is also expressed in the fetal myocardium and this correlates to the left ventricular disfunction (LVD) that is reported in 2% to 4.7% of cases with monotherapy administration [67].

The first antibody used in the treatment of patients with HER2 positive BC was trastuzumab. Trastuzumab can cause LFD of any degree in 3–7% of patients when administered as a single agent, and in 13% of patients when administered in combination with other cardiotoxic drugs [68,69].

Pertuzumab is a recombinant humanized monoclonal antibody that blocks the HER2 signaling synergically with trastuzumab in early and advanced stages of BC. In a systematic review of phase 2 and 3 randomized controlled trials testing the addiction of pertuzumab in different settings to standard therapies in patients with HER2-positive BC, pertuzumab was associated with no detectable effect on asymptomatic/minimally symptomatic LVD, but the risk of heart failure increases by approximately two-fold [70].

The CAROLE study evaluated the impact of cardiovascular disease in BC survivors previously treated with anthracycline-taxane, 5-fluorouracil, trastuzumab, pertuzumab and RT; the trial showed a high incidence of cardiovascular disease (77.6% and 51.5% in a preclinical and clinical phase respectively) and these percentages also remained higher in patients treated more than 10 years previously [71].

T-DM1 is an antibody-drug conjugate used in the metastatic setting after a treatment with pertuzumab and trastuzumab, but also in the adjuvant setting in patients with residual disease after neoadjuvant treatment with anti-HER2 double block [72]. The incidence of LVD was lower in patients treated with T-DM1 (0.8%) compared with those treated with trastuzumab plus a taxane (5.3%) [73]. The warning for cardiotoxicity is important because of prior treatment including both trastuzumab and a taxane in every setting [74]. Tyrosine-kinases inhibitors used in patients with HER2+ BC showed a minimal risk of cardiotoxicity [74]; in particular, Neratinib, approved in the adjuvant setting following trastuzumab-based therapy [75], does not carry a warning for cardiotoxicity [74].

The introduction of immune checkpoint inhibitors (ICIs) in the neoadjuvant setting for triple negative BC requests to improve the management of cardiovascular toxicity in BC patients [76]: cardiotoxicity is a very rare adverse event of ICIs, affecting up to 1% of patients, and myocarditis is the most common presentation, followed by pericarditis, vasculitis and arrythmias; the onset of cardiotoxicity is usually acute and it is not related to the duration of exposure to ICIs [77]. Despite the low percentage of myocarditis in the Keynote-522 trial (0.4% of myocarditis at any grade according to the National Cancer Institute Common Terminology Criteria for Adverse Events, version 4.0) [76], the supervision of new signs and symptoms related with ICIs-related cardiotoxicity is mandatory [77].

### 3.3. Long Term Effects of Endocrine Therapy

Because of their hormonal activity, estrogen deprivation results in a wide range of symptoms. The most frequently reported are osteoporosis and bone fractures: estrogen reduction, in fact, is associated with enhanced osteoclast activity [78]. The extension of endocrine therapy by 5 years provided no benefits in terms of disease free survival over a 2-year extension; on the other hand, it was associated with a greater risk of bone fractures [79].

Moreover, patients treated with endocrine therapy experience side effects such as genital atrophy and dizziness, weight gain, hot flashes, fatigue and musculoskeletal impairment [80].

The wide range of side effects enduring over time can cause not only a poor quality of life but also less adherence to therapy [81]. In a recent study, it has been shown that 22% of women assuming endocrine therapy discontinued it, 20% of them had a recurrence of disease, and only 11% of them completed treatment [82]. The major causes of discontinuation are associated with physical but also social well-being, and it can persist after 2 years of therapy [83].

Patients treated with AIs who reported hot flashes had a 14.2% higher 5-year discontinuation rate and a consequent shorter disease-free survival (DFS) [84]. Endocrine therapy-related hot flashes predict the higher incidence of discontinuation with but with better prognosis [84]. The use of systemic hormone replacement therapy (HRT) to mitigate symptoms related with endocrine therapy is associated with an increased risk of recurrence in patients with estrogen receptor-positive BC [85].

## 4. Sexual Disfunction after BC

### 4.1. Sex Life after BC

The diagnosis of BC has a significant impact not only on the bodies of patients but also with regard to psychological and personal concerns. Changes to the body related to the disease can contribute to a decline in self-confidence and irrational beliefs about oneself and one’s body image [86]. As a consequence, decreases in sexual frequency and satisfaction were attributed only in part to dizziness and pain during the sexual act, but also to psychological distress and changed body image [41]. The prevalence of sexual disorders is higher in BC patients than in healthy women, with percentages up to 93% [87,88,89].

The impact of psychological distress is also enhanced by endocrine therapy in the adjuvant setting: tamoxifen and AIs, in fact, can interfere with sexual performance in terms of both erotic arousal and genital changes [90]. Vulvovaginal health, directly linked to sexual health, is a key factor for female pleasure. During endocrine deprivation therapy, BC survivors are likely to present genitourinary syndrome of menopause (GSM) and sexual complaints [91]. Superficial dyspareunia is the most common symptom related with hypoestrogenism and is generally the consequence of atrophic vulvar/vaginal tissues; other symptoms are recurrent urinary tract infections, burning, discomfort, and dryness [91].

Depression and anxiety about cancer recurrence has a significant impact on sexual performance; patients can request pharmacological interventions that include drugs which act on libidinal decrease and dyspareunia, feeding the vicious cycle of sexual problematics [92,93].

Traditional solutions such as lubricants, moisturizers, and low-dose vaginal estrogen can be useful to reduce GSM symptomatology [91]. Innovative options, such as vaginal laser therapy [94] and sexual therapy are emerging as optionsto treat GSM and the sexual dysfunctions of patients [91,92].

### 4.2. Pregnancy after Breast Cancer

Due to the increasing age at first pregnancy; the number of patients with BC before the completion of the reproductive cycle is increasing [95]. In a meta-analysis evaluating 112,840 patients with breast cancer, only 6.5% of them became pregnant after diagnosis: BC survivors had a 60% reduced likelihood of having a subsequent pregnancy compared with the general population [95].

Surveys on contraceptive methods and fertility outcomes of young BC survivors and the reports about the higher use of emergency contraception compared to the general population [96] highlight the need to provide long-term contraceptive advice to women who do not wish to be pregnant [97]. An analysis from the CANTO study supports the need to raise awareness and improve the gynecological counseling for contraceptive methods in BC patients both at diagnosis and during the follow-up [98]. In addition, the CANTO trial highlights how the gynecological consultation is associated with greater contraceptive use but also with an improvement of fertility preservation strategies [98].

Several studies have investigated the impact of pregnancy after BC; there are no reported differences in oncological outcomes between BC patients with and without pregnancy [99]. In the same way, no differences in terms of DFS were observed between patients who became pregnant 2 years before or after BC diagnosis [100].

In patients with a germline BRCA mutation, pregnancy after BC is safe without worsening maternal oncological prognosis or fetal problems [101]. Despite the controversial role of ovarian suppression with gonadotropin-releasing hormone agonists, its efficacy and safety during chemotherapy seem to be an available option to reduce premature ovarian insufficiency and increase the likelihood of fertility in young BC patients [102].

No differences in reproductive outcomes, such as risk of congenital abnormalities, were observed [95]. Women treated for BC are more at risk of obstetrical issues; the incidence of low birth weight, fetal disorders, preterm birth and low size of the baby for gestational age is higher in women who underwent treatment for BC [103,104].

Many young patients of reproductive age have not experienced maternity at the time of diagnosis, making treatment-associated infertility one of the most significant problems. This evidence enounces the necessity of oncofertility counseling in our patients during reproductive age before the start of any gonadotoxic therapies [105].

Oncofertility counselling requires a multi-specialist approach with the involvement of oncologists, surgeons, endocrinologists, gynecologists, psychologists and reproductive medicine specialists, and should be customized based on patient and disease/treatment-related factors [105]. Nevertheless, a low percentage (9–14%) of patients received adequate information about their reproductive potential [106,107,108]. It is important to define and implement a standardized reproductive counseling protocol and to propose and encourage counseling even when patients do not desire/cannot yet have a pregnancy, to have a tailored and long-term evaluation about the potential risk of fertility reduction after BC treatment [109].

Available strategies for ovarian function include controlled ovarian stimulation (COS) and the cryopreservation of oocytes and ovarian tissue and the administration of gonadotropin-releasing hormone agonists (GnRHa) during cytotoxic therapy [108]: Table 2 summarizes the most important fertility preservation techniques used for BC patients. COS protocols are safe for the outcome of BC [110] and different trials are evaluating the impact of random-start ovarian stimulation to reduce the time needed prior to the oncological treatment [111].

Despite the known evidence about the protective role of breastfeeding the prevention of BC, the literature about the management of lactation after BC is limited [115].

Milk volume can be reduced in patients with a BC diagnosis, not only after mastectomy but also with conservative surgery: milk production, in fact, can be reduced because of the sequelae of surgery, such as ductal obstructions or nipple trauma, but also because of adjuvant treatments [115]. In addition, irradiation of the breast causes histological changes such as atypia, fibrosis and vascular abnormalities in the breast that can compromise milk production [116].

On the other hand, some adjuvant treatments are contraindicated during breastfeeding: the use of AIs is forbidden because they pass into the milk and can inhibit the production of estrogens in the infant. Instead, the bioavailability of tamoxifen in the milk is not known [115].

## 5. Unmet Clinical Needs

### 5.1. Body Weight Control and Lifestyle Intervention

The nutritional intervention in patients with BC has significant clinical relevance in every phase of the disease. During chemotherapy, it helps to have an adequate intake of energy reducing toxicity, but also during the follow up, habit changes, including diet and exercise, help to guarantee a healthy lifestyle and to control BC comorbidities [117].

Basal body weight has emerged as an important prognostic factor in patients with BC: obese patients, in fact, had a worse prognosis and a significantly increased risk of recurrence [118]; in patients with luminal BC this influence is bigger [119].

Endocrine adjuvant therapy, in fact, can modify the body composition. AIs reduce aromatases inducing insulin resistance and, as a consequence, patients treated with AIs had a greater rate of body fat and insulin resistance [120].

The LEAN study is a randomized trial investigating the role of physical intervention on BC survivors. Patients randomized in the intervention group had to reduce the intake of fat and to increase fruit and fiber. Patients who had lost ≥ 5% of body weight had a better healthy eating index score [121]. The benefits of improvement of lifestyle are also associated with physical activity; aerobic exercise improved quality of life and fitness in BC survivors [122].

In a prospective analysis, fruit and vegetable consumption was investigated among women diagnosed with BC, and it was found that vegetable consumption in particular was correlated to better survival, although not breast-cancer specific survival [123].

On the other hand, nutrition science usually shows imprecise results due to the difficulties of clinical trials in this field. For this reason, the suggestion of a “healthy” dietary pattern without specific foodstuffs or components, and the encouraged improvement of their lifestyle is part of the therapeutic approach in BC patients [117].

### 5.2. Psychological Implications of Breast Cancer and Pathogenetic Germinal Mutations

A BC diagnosis represents a stressful event for the patient and their family. Accepting the diagnosis, undergoing treatments with their possible side effects, understanding the prognosis, and accepting an uncertain future are all stages that can cause psychological instability and that enhance the incidence of anxiety and depression in BC patients [124].

Emotional distress brings to a reduction in quality of life among patients with a negative impact on compliance treatment and a consequent elevated risk of mortality; these evidences suggest evaluating distress as a vital sign in cancer patients [125].

A systematic review found a prevalence of 9.4–66.1% for depression and 17.9–33.3% for anxiety among BC survivors [126]. An interesting cross-sectional study investigated the prevalence of symptoms of depression and/or anxiety between 350 BC survivors treated with chemo- and/or radiotherapy five years before and 350 women of the same age without cancer, 3.7% of patients with a diagnosis of BC had severe symptoms of depression compared to 1.1% of controls. Similar results were found for anxiety, which was reported in 8% of BC survivors and 4% of controls [127].

Cancer has many sequelae, such as the inability to see a future, coping with a new body and life, and one of the main causes of psychological distress is the fear of recurrence [128]. Many factors are associated with the fear of recurrence, such as age; younger patients have greater stress than older ones [129]. Interestingly, it has been supposed that breast-conserving therapy could positively influence the lives of cancer patients, but if effective in the short-term, it doesn’t seem to persist in the long-term because some studies show how patients who underwent a mastectomy have less fear of recurrence than ones who underwent breast-conserving therapy [129,130].

Offering the proper treatment for psychological well-being is a fundamental issue to prevent negative emotional outcomes. Different psychological interventions can be utilized: a meta-analysis showed how cognitive behavioral therapy seems to reduce symptoms of depression and anxiety [131]. The cognitive-behavioral therapy is a structured approach that aims to solve problems by modifying behavior and unhelpful thinking [132]. Mindfulness-based stress reduction has been shown to have a positive influence on stress reported by patients, also resulting in low salivary cortisol and IL-6 in breast cancer survivors [133].

BC remains an experience with long-term effects on the well-being of patients, but clinicians have an important role to play as part of the disease, helping the patient to reprocess the experience of cancer, which could be a key to promoting resilience among survivors [134].

The diagnosis of a germinal mutation in genes involved in the hereditary predisposition to breast and ovarian cancer is often related with anxiety and depression, constant worries about cancer recurrence and an important impact on social and psychological factors [135]. In addition, counseling after the diagnosis of the pathogenic variant includes discussions about the preventive surgery to reduce the risk, the management reproductive desires and menopausal symptoms, and the evaluation of quality-of-life aspects related with risk-reducing surgical procedures [136].

Van Oostrom et al. evaluated the long term psychological impact in the mutation carriers. They did not differ from non-carriers BC survivors showing a significant increase in anxiety and depression in the first 5 years of follow-up. BRCA carriers having undergone prophylactic surgery had a significant benefit in terms of reduction of fear of developing cancer but, at the same time, they reported a body image changes with a less favorable impact on sexual relationship (70% of cases). Distress levels increased if patients had young children or had lost a relative to cancer [137].

## 6. Conclusions

Because of the improvement of the efficacy of treatments and the reduction of cancer-related mortality, the prevalence of BC survivors has increased in recent years. This leads oncologists and patients to face different early and late issues related to BC treatments.

BC survivors can experience different adverse events related to BC treatment related to local or systemic treatment that have different implications on the psychological and gynecological spheres.

One of the most important challenges for oncologists is to know and, if possible, prevent the onset of adverse events related with BC diagnosis and treatments that can afflict and worsen the quality of life of patients.

## Figures and Tables

**Figure 1 jcm-11-07239-f001:**
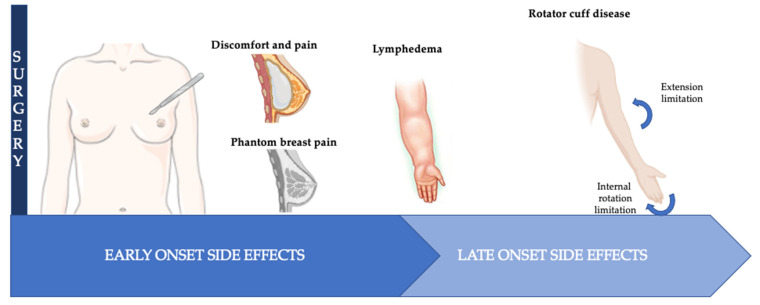
Most important side effects of surgery and timing of onset.

**Table 1 jcm-11-07239-t001:** Ongoing trial evaluating the impact of omission/reduction of RT for BC patients.

Trial Name	Main Inclusion Criteria	Profiling Risk	Study Design	Ref.
PRECISION trial (NCT02653755)	50–75 years old Conserving-surgery for pT1 N0 ER+ (≥10%) or PR+, HER2- and G1/2	PAM50 transcriptional profile used for risk stratification.	Non randomized phase II trial	[33]
NCT03646955	Age ≥ 60 years Conserving-surgery for pT1 N0 ER ≥10% HER2- non-lobular G1/2	Clinical risk stratification	Randomized phase III trial	[34]
NCT05371860	Age ≥ 40 years HER2 positive early BC cT1-3, N0, M0 before NACT and targeted HER2 therapy Documented pCR after NACT	Clinical risk stratification	Non randomized trial	[35]
DESCARTES trial (NCT05416164)	Age ≥ 18 years ER positive/HER2 negative, HER2+ (ER/PR +/−) or TNBC cT1-2, N0, M0 before NACT Documented pCR after NACT	Clinical risk stratification	Non randomized trial	[36]
**NCT04517266**	Age ≥ 18 years high risk BC (≥2 clinical risk factors) p T > 1, N1 Radical or conserving surgery and ALND	Clinical risk stratification to omit the irradiation of internal mammary nodes	Randomized trial	[37]

Abbreviations: ER: estrogen receptor, PR: progesterone receptor, HER2: epidermal growth receptor 2; G1/2: grade 1 or 2; PAM50: prediction analyses of microarray 50; NACT: neo-adjuvant chemotherapy, pCR: pathological complete response; TNBC: triple negative breast cancer; ALND: axillary lymph node dissection.

**Table 2 jcm-11-07239-t002:** Fertility preservation options for BC patients prior to and during anticancer treatment.

Technique	Procedure	Risk Related with Procedure	Ref.
COS and cryopreservation of oocyte	Ovarian stimulation with estrogen, oocyte pick up and egg banking.	Exposition to estrogen during the induction	[112]
	Required approximately 2–4 weeks prior to starting any cancer treatment	Ovarian hyperstimulation syndrome	[113]
COS and cryopreservation of embryos	Ovarian stimulation with estrogen, oocyte pick up, in vitro fertilization (requests a sperm donor) and embryo banking.	Exposition to estrogen during the induction	[112]
	Required approximately 2–4 weeks prior to starting any cancer treatment	Ovarian hyperstimulation syndrome	[113]
Ovarian tissue cryopreservation	Ovarian tissue surgically removed, cryopreserved and reimplanted after cancer treatment	Performed only in selected centers	[112]
	No daley in starting cancer treatment	Requires surgery	[113]
GnRHa administration	Hormonal treatment administrated during the full duration of anticancer treatment	Controvertial and debated role	[114]

Abbreviations: COS: controlled ovarian stimulation; GnRHa: gonadotropin-releasing hormone agonists.

## Data Availability

Not applicable.

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
