# Peer review of "Long-Term Effects of Breast Cancer Therapy and Care: Calm after the Storm?"

_jcm, 2022, doi:10.3390/jcm11237239_

Round 1

Reviewer 1 Report

The Review by Tommasi and colleagues provides an excellent overview of the long-term effects of breast cancer therapy and care.  The topics covered range from surgery and radiotherapy to lymphedema and neuropathy - - with a special emphasis on the psychological consequences of the diagnosis and the impact on quality of life issues.  The paper is basically well written - - with the following recommended revisions:

- Line 44: “685000” should be changed to “685,000”

- Lines 46/47:  Could the authors comment about breast cancer survival worldwide?

- Line 72:  Change “of” to “with”

- Line 78:  Change “As regards” to “Regarding”

- Line 80:  Change “shoulders” to “shoulder”

- Line 87:  Change “lead” to “leads”

- Line 88:  Change “current” to “currently”

- Line 125:  Delete “up”

- Line 128:  Change “radiations” to “radiation”

- Line 133:  Change “trial” to “trials”

- Line 142:  Change “has” to “have”

- Line 145:  Change “As regards” to “Regarding”

- Line 206:  Add “the” before “cardiovascular”

- Line 208:  Add “a” before “known”

- Line 217:  Change “antibody” to “antibodies”

- Line 218:  Change “it is” to “they are”

- Line 242:  Add (,) after “pericarditis”

- Line 243:  Delete “and” before vasculitis; change “od” to “of”

- Line 255:  Change “side-effected” to side effects

- Line 284:  Add “such” before “as”

- Line 288:  Change “112.840” to “112,840”

- Line 387:  Change “challenge” to “challenges”

This paper will be of special interest to clinicians, researchers, and breast cancer survivors.

Author Response

Thanks for the precious and accurate suggestions. 

I corrected the reported revisions.

Reviewer 2 Report

Thank you for requesting to provide a review of this article, which has a subject of high interest, as breast cancer continues to be the most common cancer among women and the leading cause of death in women. 

               The main aim of the review was to summarize the consequences of medical and surgical treatment on breast cancer patients, to emphasize the importance of early prevention of side effects and to improve the life quality of the patient’s.

              Regarding the structure and accuracy of the phrases, the manuscript has well structured information, and the phrases are well understandable.

             The manuscript is original and well defined. 

             The results provide an advance in current knowledge. 

             The results are being interpreted appropriately and are significant. 

             The data are robust enough to draw the conclusion.

             Surely the paper will attract a wide readership. 

             The English language is appropriate and has only a few writting mistakes, which need to be corrected so that the article can be of highest quality.

             There are only a few things to be added in the lines below:

Line 49: the paragraph that begins with „It is already...” should be aligned with the previous paragraph

Line 51: without „.” between „treatments” and „[3]”

Line 54: the paragraph that begins with „BC survivors...” should be aligned with the previous paragraph

Line 57: „,” before „but”

Line 59: the paragraph that begins with „The aim...” should be aligned with the previous paragraph

Line 72: are diagnosed with, not „are diagnosed of”

Line 87: which leads, not „which lead”

Line 240: 1 space required between „(ICIs)” and „in”.

Author Response

Thanks for the precious suggestions and the revisions. 

I fixed reported errors and other during the revision.